# Complex Material and Surface Analysis of Anterolateral Distal Tibial Plate of 1.4441 Steel

**Josef Hlinka** [1,2,*] , **Kamila Dostalova** [2], **Katerina Peterek Dedkova** [2], **Roman Madeja** [3,4], **Karel Frydrysek** [4] , **Jan Koutecky** [5], **Pavel Sova** [1,2] and **Timothy E. L. Douglas** [6,7]

1 Department of Materials Engineering, Faculty of Materials and Technology, VSB-Technical University of Ostrava, 17. Listopadu 2172/15, 708 00 Ostrava-Poruba, Czech Republic; pavel.sova@vsb.cz
2 Centre for Advanced Innovation Technologies, VSB-Technical University of Ostrava, 17. Listopadu 2172/15, 708 00 Ostrava-Poruba, Czech Republic; kamila.dostalova@vsb.cz (K.D.); katerina.dedkova@vsb.cz (K.P.D.)
3 Trauma Center, University Hospital Ostrava, 17. Listopadu 1790, 708 52 Ostrava-Poruba, Czech Republic; roman.madeja@fno.cz
4 Institute of Emengency Medicine, University of Ostrava, Syllabova 19, 703 00 Ostrava, Czech Republic; karel.frydrysek@vsb.cz
5 Medin a.s., Vlachovicka 619, 592 31 Nove Mesto na Morave, Czech Republic; jan.koutecky@medin.cz
6 Engineering Department, Gillow Avenue, Lancaster University, Lancaster LA1 4YW, UK; t.douglas@lancaster.ac.uk
7 Materials Science Institute (MSI), Lancaster University, Lancaster LA1 4YW, UK
* Correspondence: josef.hlinka@vsb.cz

**Abstract:** Nickel-based austenitic stainless steels are still common for manufacture of implants intended for acute hard tissue reinforcement or stabilization, but the risk of negative reactions due to soluble nickel-rich corrosion products must be considered seriously. Corrosion processes may even be accelerated by the evolution of microstructure caused by excessive heat during machining, etc. Therefore, this study also deals with the investigation of microstructure and microhardness changes near the threaded holes of the anterolateral distal tibial plate containing approx. 14wt.% Ni by composition. There were only insignificant changes of microhardness, grain size, or microstructure orientation found close to the area of machining. In addition, wettability measurements of surface energy demonstrated only minor differences for bulk material and areas close to machining. The cyclic potentiodynamic polarization tests were performed in isotonic physiological solution. The first cycle was used for the determination of corrosion characteristics of the implant after chemical passivation, the second cycle was used to simulate real material behavior under the condition of previous surface damage by excessive pitting corrosion occurring during previous polarization. It was found that the damaged and spontaneously repassived surface showed a three-time higher standard corrosion rate than the "as received" chemically passivated surface. One may conclude that previous surface damage may decrease the lifetime of the implant significantly and increase the amount of nickel-based corrosion products distributed into surrounding tissues.

**Keywords:** pitting corrosion; microstructure; implant; traumatology; cytotoxicity; surface contact angle; chemical passivation

## 1. Introduction

Metallic materials are widely used in a large number of implantology applications. Although there are benefits, some complications can occur after the insertion of a metal-based implant into the body. These complications can be classified according to their origin. Some are caused by ill-considered construction design (involving shape and size). Others can be caused by inappropriate material selection. Currently, there is an effort to avoid problems with construction design mainly by a custom-made approach, especially in cases where the surgical treatment can be planned. The topic of inappropriate material selection is more complicated because of the sensitivity of the body to some elements. A typical

metallic implant is a solid object that is implanted into a human body during surgery. Even though research in the field of implant production is continuing with the tendency to find materials with properties nearly similar to those of bones, implants are still artificial objects for the body. Therefore, chronic inflammations in the implantation site or allergic reactions can occur. In some cases, the body has a tendency to eliminate the implant. There are a couple of ways to prevent this consequence. The implant surface is the main interface between the material of the implant and the human body environment. Hence, it can be modified in various ways to imitate natural body structure. Generally, the implant surface is covered with a layer that is tolerable for a body and its character is given by its roughness, wettability, chemical composition, etc. There is always a risk of scratching the layer during implant implantation. Moreover, implants commonly used for long-term fixation or reinforcement of damaged hard tissues are typically repetitively stressed by axial and uniaxial forces which may result in premature development of fatigue cracks in their structure [1]. Moreover, when the implant is in direct continuous contact with other moving parts or tissues, wear damage can appear due to undesirable friction between these parts if any movement or instability occurs [2]. These mechanisms always have to be considered, even if they do not occur in every case. As the human body contains water-based liquids, all degradation processes related to mechanical actions are synergistically accelerated by corrosion processes [3,4]. The character of an implant surface establishes not only corrosion resistance but also determines the biological response of the tissue [5]. These aspects make corrosion and technological properties of implants crucial for their proper design, modeling, and determination of their lifecycle. This research focuses on the investigation of the surface properties (microstructure, corrosion, wettability, microhardness, and contact-type surface roughness test) of an implant intended for ankle reparation consisting of AISI 1.4441. As this nickel-rich austenitic steel is widely used for manufacturing mainly short-term hard tissue reinforcements, complications related to ions releasing during specific corrosion processes may result in its terminal failure. The motivation of the research presented here is to primarily evaluate the amount of ions released from the application under conditions simulating its real use.

## 2. Materials and Methods

An AISI 1.4441 steel was used for manufacture of anterolateral distal tibial plates which was further tested by experimental techniques (Figure 1). It is an L-shaped board, anatomically shaped for the left and right sides in lengths of 2 rotations (90 mm) to 16 rotations (300 mm). There are holes in the plate for the angularly stable screws of 3.5 mm, possibly for 3.5/2.7 mm screws and K-wires of Ø 1.5 mm. Three screw holes are divergently directed in the distal part; the other two pairs of screws form their support and fix the fragments in a skew line to the distal screws. The distal part includes a protrusion allowing the fixation of a possibly broken Chaput tubercle. The holes are positioned in the shaft part of the plate, perpendicular to the plate. The anterolateral distal tibial plate is intended for osteosynthesis of complicated fractures of the distal tibia, some storey fractures, or fractures of the distal tibia with a fracture line in the frontal plane. The standardized chemical composition of the studied material is listed in Table 1. The surface of the plate was mechanically polished and pickled in a mixture of hydrofluoric and nitric acid to remove any free particles or to degrade and eliminate any grease and oil residues from previous manufacturing processes. The last step before repeated ultrasonic cleaning in distilled water was surface passivation with nitric acid at a concentration of 30% at a temperature of 50–60 °C for 20 min. The passivation prescription is in accordance with ASTM A967, widely considered as a standard for this type of material [6].

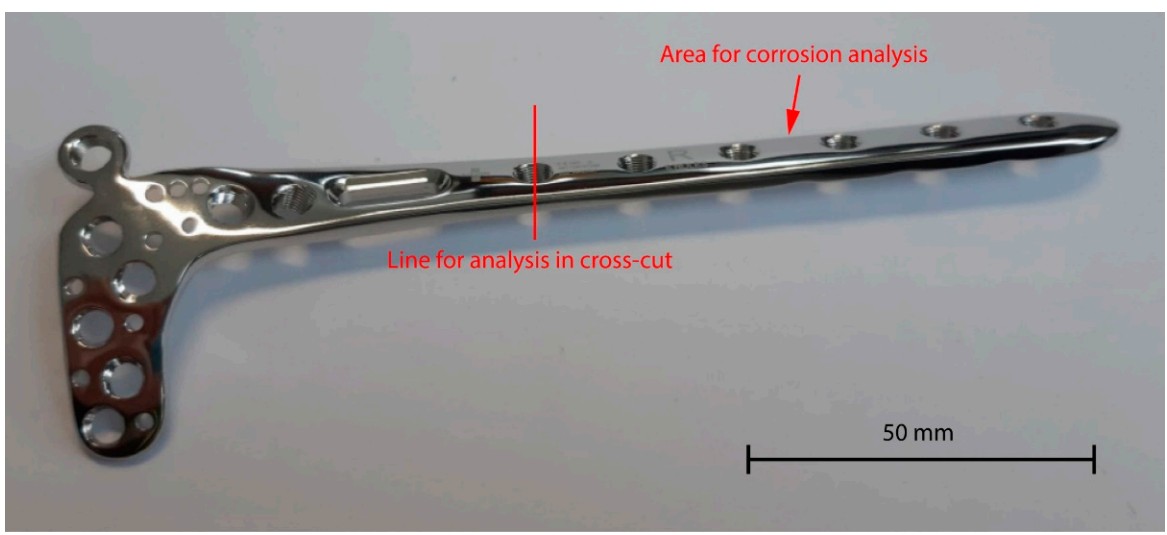

**Figure 1.** Anterolateral distal tibial plate with marked areas of analysis.

**Table 1.** Standardized chemical composition of 1.4441 steel according to [7].

| AISI 1.4441, Chemical Composition (wt.%) | | | | | | | | |
|---|---|---|---|---|---|---|---|---|
| **C** | **Si** | **Mn** | **P** | **S** | **Cr** | **Mo** | **Ni** | **Others** |
| <0.03 | <1 | <2 | <0.25 | <0.01 | 17–19 | 2.5–3.2 | 13–15 | N < 0.1, Cu < 0.5 |

### 2.1. Microstructure and Metallography Observation

For the evaluation of the corrosive effect and the basic semi-quantitative chemical properties, an analysis of the surface layer was performed using an SEM FEI 450 Quanta FEG (FEI Company, Brno, Czech Republic) equipped with an EDAX EDS detector (AMATEK Company, Tilburg, The Netherlands) in the secondary electron mode. Accelerating the voltage to 15 keV enabled analysis of a wide range of chemical elements from the periodic table. Due to the shape of the analyzed sample, the working distance was 10–15 mm. The samples for metallography and microhardness testing were mounted into bakelite resin (Polyfast) with carbon particles filler supplied by Struers (Roztoky, Czech Republic). This resin stabilizes the samples during mechanical preparation and microhardness testing safely. The metallography observations were performed on samples after mechanical polishing using equipment and diamond suspensions made by Struers (Roztoky, Czech Republic) with chemical etching (22 °C/60 s) in a modified Vilella's reagent [7] containing 10 parts 35% HCl, 10 parts distilled $H_2O$, and 1 part 65% $HNO_3$. The image capturing and evaluation was performed by an Olympus IX70 inverted metallographic microscope (Olympus, Prague, Czech Republic).

### 2.2. Corrosion Test

Due to a very low corrosion rate of AISI 1.4441 under standard conditions, the corrosion characteristic has to be obtained by accelerated corrosion tests. Potentiodynamic polarization tests were performed in high-density polyethylene and high-density polypropylene corrosion cells with lower exposure hole which exposes 0.49 $cm^2$ of the tested material. A similar corrosion cell setup is often used when complex surfaces are evaluated [8]. A hardware device, Voltalab PGZ 100 with Voltamaster 10 software (Villeurbanne, France), with 3 electrode setups was used. The testing method followed ASTM F 2129, ASTM G 61, and ISO 12,732 with certain temperature and gas bubbling modifications regarding subsequent application in biomedical engineering. This setup allows bubbles formed on the surface during tests to escape freely and not to affect the continuity of the measurements. A three-electrode setup was used for precise measurement. The sample was connected as

a working electrode, saturated calomel electrode (SCE, +241 mV vs. Saturated Hydrogen Electrode (SHE)) [9] served as a reference electrode, and a high purity carbon rod was connected as an auxiliary electrode. The physiological saline solution (0.9 wt.% NaCl in distilled $H_2O$) was used as a corrosion solution for potentiodynamic polarization to intentionally simulate the environment of living tissue. The testing temperature was standard: 25 °C.

There was a 60 min time gap applied after filling corrosion cells with a physiological solution to stabilize partial corrosion processes. Before starting the potentiodynamic polarization, the initial potential value was set to −80 mV vs. the potential after stabilization of the corrosion equilibrium (OCP), with the polarization rate set to 60 mV·min$^{-1}$ [10]. The dependence of the current flowing through the potential applied to the test sample was recorded during the measurement. The potential was gradually applied to the tested sample, which increased over time with the value of the polarization rate. There were two polarization curves measured in this experiment. The first curve represents the corrosion behavior of the chemically passivated surface and the second curve was recorded to illustrate materials' self-passivation abilities in physiological solution. Once more, a 60 min time gap was applied between the two polarizations. Each polarization test was terminated when the value of corrosion current density reached $2 \times 10^{-3} A/cm^{-2}$, which ensured that the material was located in a transpassive state and the surface was actively corroded [11].

### 2.3. Wettability Test

The surface angle between the sample and water was evaluated by the sessile drop method. The surface contact angle was found by the SEE (surface energy evaluation) system and free surface energy was calculated by Advex Instrument software (Brno, Czech Republic). We applied 3 µL droplets of high purity water to the tested surface and the contact angle θ was determined by the tangent to the drop profile at the point of contact of the three phases (liquid, solid, gas) with the plane of the sample surface [12]. The free surface energy of the solid sample is determined by Young's Equation (1), where $\gamma_{SV}$, $\gamma_{LV}$, and $\gamma_{SL}$ represent the interfacial tensions per unit length of the solid-vapor, liquid-vapor, and solid-liquid contact line, respectively [13].

$$\gamma_{SV} - \gamma_{SL} = \gamma_{LV} \times \cos \theta \qquad (1)$$

### 2.4. Microhardness Testing

Metallographic samples were tested for microhardness repeatedly. The smooth surface after diamond paste polishing allows low loading force hardness testing to be performed, so only HV 0.1 (1 N, approx. 0.1 kg) could be used to determine hardness parameters of the implant microstructure. By this method, the hardness of separate grains in microstructure could be measured easily. The test was performed according to ASTM E92 using a LECO AMH 2000 (Plzen, Czech Republic) equipped with a diamond Vicker's indentor and a high-resolution camera. The Vickers microhardness can be calculated by Equation (2), where F is the value of applied loading force in N and $d_1$ and $d_2$ are diagonals of studied indent [14].

$$HV (F) = 0.189 \times F/[((d_1 + d_2)/2)]^2 \qquad (2)$$

### 2.5. Contact-Type Surface Roughness Measurement

The surface roughness of the machined workpieces was measured using the conventional stylus instrument. A Taylor-Hobson Talysurf Intra 50 profilometer was used to measure the average roughness value (Ra) and average maximal height of profile (Rz) [15]. The device was equipped with a floating arm with a diamond tip of a radius of 2 µm. A 1 mm/s canning rate was set to follow ASTM D7127 standard. The measured length was set to 5 mm due to the complicated geometrical shape of the surface to be characterized.

## 3. Results

### 3.1. Microstructure and Metallography Observation

Figure 2A shows a longitudinal cut near a threaded hole. The microstructure near the thread contains equiaxial austenitic grains with no signs of carbide and oxide particles, neither inside of grains nor at the grain boundaries, and fully corresponds with the microstructure of the bulk material presented in Figure 2B. The microstructure near the threaded hole also shows no changes in texture, grain shape, or size. This area is crucial due to its role as a stress concentrator during force loading and transmitting into the rest of the application so any microstructural changes may cause cracks formation leading to premature failure. The machining cut was precisely driven through the grains which are indicated by the smooth interface between the threaded hole and mounting resin with no significant deformation zone visible in the materials' microstructure [16].

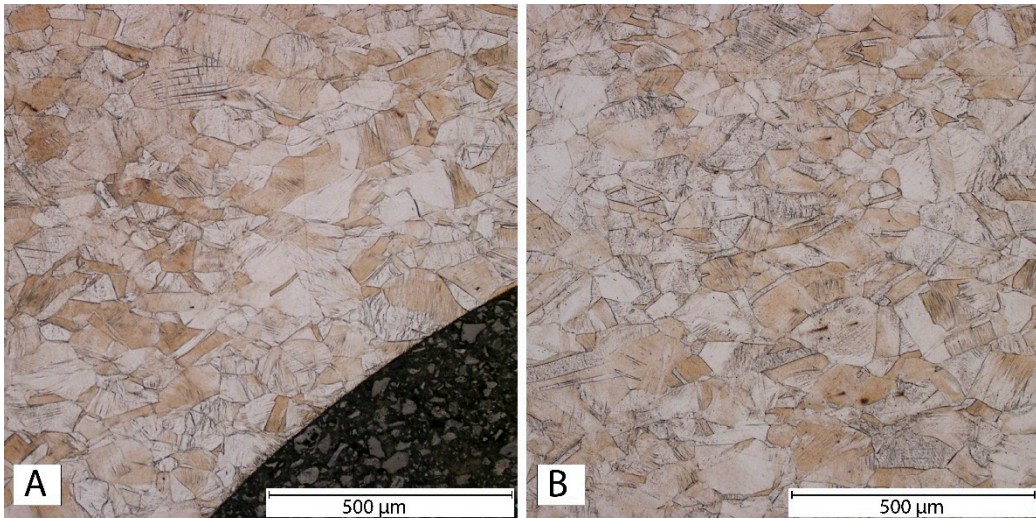

**Figure 2.** (**A**) Microstructure of material near the threaded hole, (**B**) microstructure of bulk material.

### 3.2. Microhardness Testing

The microhardness of the samples was tested and evaluated after polishing so the diagonals of indents could be measured precisely. There was a line testing set into device software to evaluate microhardness changes from the surface into the bulk material. The zero position was set at a distance of 100 μm from the interface between the threaded hole and mounting resin and the step between each indent was set to 200 μm for both longitudinal transversal cuts. The values of each measurement are listed in Table 2 together with average and standard deviation values.

The average microhardness is increased in comparison to previously published values [17]. As the difference between each value is significant and standard deviations are approximately 4% for the longitudinal and 6% for the transversal direction compared to average microhardness, the microstructure near the indents was further investigated. Figure 3A illustrates indents in a longitudinal cut after polishing and Figure 3B after etching where the lower magnification of microscope was used to capture more indents. Significant differences in microstructure were observed for each indent. The smallest indents (highest microhardness) were measured in grains showing a high level of deformation indicated by the presence of deformation twins [18]. Some high values also indicate the possibility of initiation of ε martensite transformation [19].

**Table 2.** Values of microhardness HV 0.1 for transversal and longitudinal cut direction.

| Longitudinal Direction | | Transversal Direction | |
|---|---|---|---|
| Distance from the Thread (mm) | HV 0.1 | Distance from the Thread (mm) | HV 0.1 |
| 0.1 | 346 | 0.1 | 321 |
| 0.3 | 327 | 0.3 | 304 |
| 0.5 | 302 | 0.5 | 346 |
| 0.7 | 338 | 0.7 | 304 |
| 0.9 | 338 | 0.9 | 295 |
| 1.1 | 332 | 1.1 | 331 |
| 1.3 | 308 | 1.3 | 361 |
| 1.5 | 327 | 1.5 | 314 |
| 1.7 | 327 | 1.7 | 294 |
| 1.9 | 333 | 1.9 | 301 |
| Average HV 0.1 | 328 | Average HV 0.1 | 317 |
| Standard deviation | 14 | Standard deviation | 20 |

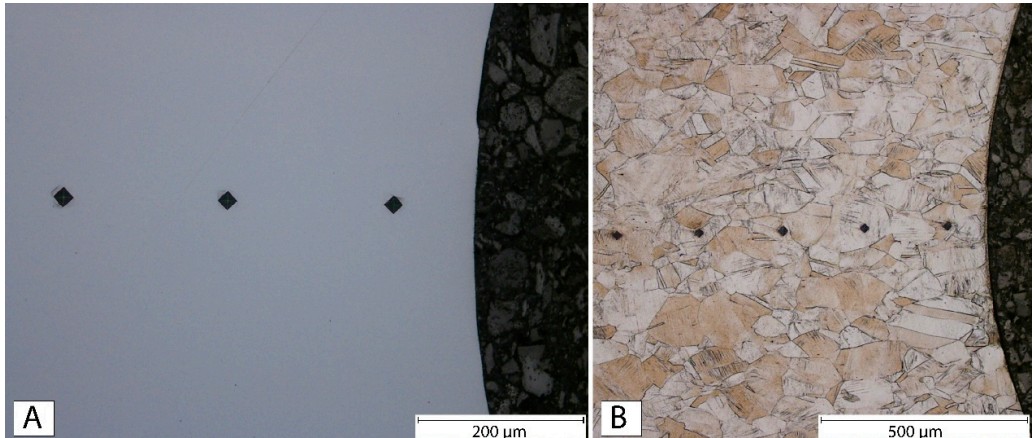

**Figure 3.** (**A**) Indents after polishing, (**B**) identic indents in the revealed microstructure.

*3.3. Wettability Test*

The polished surface was used for the wettability test to avoid any effect of local roughness or unevenness on contact angle values [20]. Before the test, the sample was cleaned in an ultrasonic bath firstly in acetone, then in double distilled water with the testing surface facing up to avoid being scratched. Average values of contact angle and calculated surface energy and their standard deviations are presented in Table 3. The standard deviation of the presented surface energy is not symmetrical due to the cosine function used for its calculation. Representative images of droplets on tested surfaces are shown in Figure 4.

**Table 3.** Value of measured contact angle and calculated surface energy.

| Sample | Contact Angle (°) | Surface Energy (mJ·m$^{-2}$) |
|---|---|---|
| Close thread | $50 \pm 4$ | 53.5 + 2.6; −2.3 |
| Between threads | $52 \pm 2$ | 53.6 + 1.2; −1.2 |

*3.4. Corrosion Testing Methods*

There were two polarization curves collected for the purpose of precise investigation of the tested application corrosion properties. The testing was repeated at the same location with no change of testing instrument. The first polarization curve represents the electrochemical properties after surface finishing procedures (machining, grinding,

polishing, degreasing, chemical passivation, sterilization, etc.). The second curve illustrates the behavior of the same material as the previous polarization actively removed more thermodynamically active system elements (thick passive layer, secondary phases particles, oxide layers, deformed material layer, etc.) after the spontaneous formation of the passive layer in physiological solution on the previously corroded surface. Values of current density (Y-axis) and potential (X-axis) were continuously recorded during polarization. After the polarization procedure, the semilogarithmic polarization curve was drawn up from these points and is illustrated in Figure 5. Finally, the corrosion properties (corrosion potentials, polarization resistance corrosion current density, and a corrosion rate) were calculated from the initial part of the polarization curves with the characteristic "V-shape" by Tafel extrapolation automatically using Volta Master 10 software [21]. These are listed in Table 4. There was an exchange of two electrons ($Fe^0{\rightarrow}Fe^{2+}$, $Ni^0{\rightarrow}Ni^{2+}$, $Cr^0{\rightarrow}Cr^{3+}$), and an average material molar mass of 56.2 g/mol and density of 7.8 g/cm$^3$ was considered for calculation of corrosion rate by Faraday laws [22].

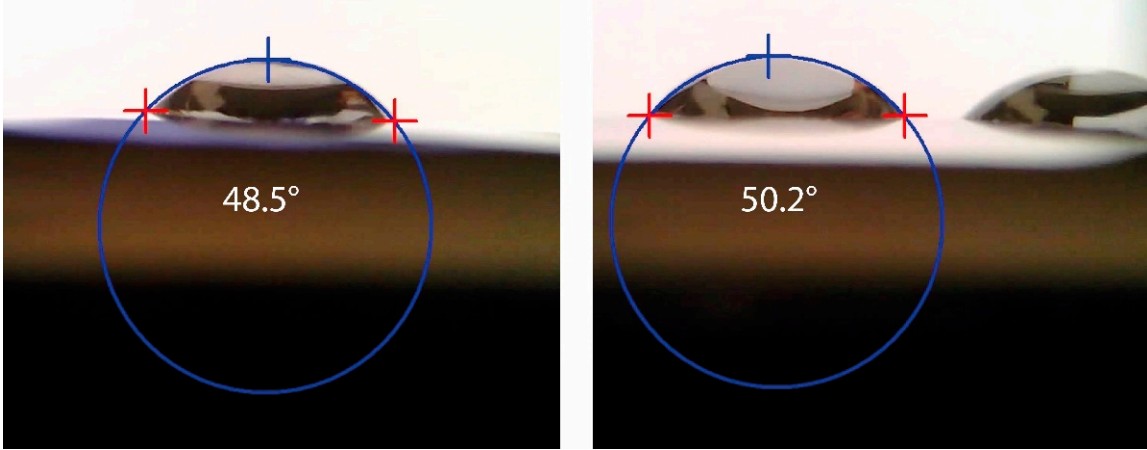

**Figure 4.** Water droplets on the surface of the tested sample. The value of contact angle evaluated is given.

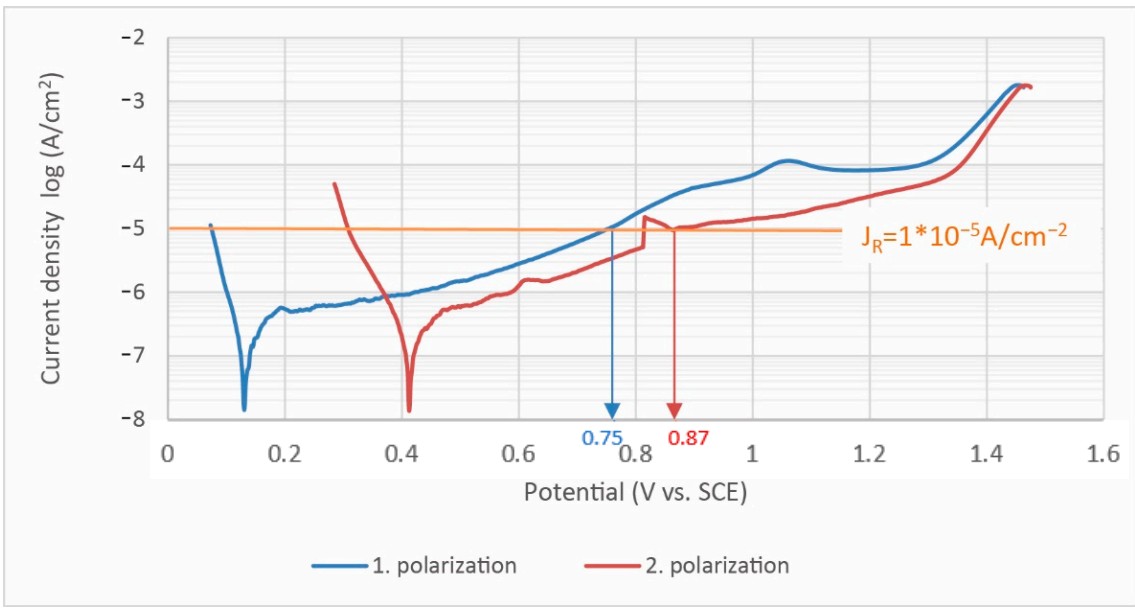

**Figure 5.** Polarization curves of 1.4441 for first and second polarization in physiological solution.

**Table 4.** Corrosion properties obtained by Tafel extrapolation.

| Curve NO. | Corrosion Potential Ecor (mV vs. SCE) | Corrosion Rate $C_R$ (µm/Year) | Polarization Resistance Rp (kΩ·cm²) | Corrosion Current Density JC (µA/cm²) |
|---|---|---|---|---|
| 1 | 134 | 0.76 | 107 | 0.065 |
| 2 | 414 | 2.01 | 84 | 0.172 |

As the anodic part of the curve exhibits no inflection point, the pitting potentials represented by orange and blue arrows were determined from the critical value of current density 10 µA/cm², which is given by the dotted green line in Figure 5. This potential is formally defined by destabilization of the passive layer or formation and spontaneous growth of corrosion pits, respectively. The values for each curve are marked on the potential axis.

As there are metal ions released from the material during the spontaneous corrosion process, the weight of released ions during the time interval from defined surfaces can be theoretically calculated from the chemical composition of the steel and values of corrosion current density by modification of Faraday's law of electrolysis (1). The following calculation is shown for calculation of the mass of released ions during a year period.

$$m^z + = (J_C \times t \times M \times X)/(F \times z) \tag{3}$$

where:

- m is the mass of ions released during corrosion in grams,
- $J_C$ is corrosion current density,
- t is time in seconds (31,536,000 s/year),
- M is the molar mass of the substance in grams per mol,
- X is the atomic volume fraction of metal in steel composition (e.g., 0.15 for Ni in 1.4441 steel),
- F is the Faraday constant (96.485 Coulomb per mol), andS
- z is the valency number of ions (electrons transferred per ion during the reaction).

Equation (1) was used for calculation of theoretical ion release from 1 cm² of material during its 1-year period of exposure under conditions used for corrosion test. Table 1 provides the maximal atomic volume fraction of manganese, chromium, molybdenum, and nickel which were considered for the calculations. Calculated results are given in Table 5, which also shows the results after recalculation to the amount of substance of released ions.

**Table 5.** Theoretical mass and amount of ions released from 1 cm² during the 1-year exposition.

| Polarization | µg/(Year·cm²) | | | |
|---|---|---|---|---|
| | **Mn** | **Cr** | **Mo** | **Ni** |
| 1st polarization | 12.5 | 118.3 | 19.9 | 99.6 |
| 2nd polarization | 32.9 | 312.6 | 52.6 | 263.2 |

| Polarization | µmol/(Year·cm²) | | | |
|---|---|---|---|---|
| | **Mn** | **Cr** | **Mo** | **Ni** |
| 1st polarization | 0.2 | 2.3 | 0.2 | 1.7 |
| 2nd polarization | 0.6 | 6.0 | 0.5 | 4.5 |

After the corrosion test was terminated, the area of interest was observed using a scanning electron microscope (SEM, FEI QANTA FEG 450, Brno, Czechia), in a regime of secondary (SE) and backscattered (BSE) electrons. The semiquantitative chemical analysis of marked areas was performed by energy-dispersive X-ray spectroscopy analysis (EDX). The result of the observation is illustrated in Figure 6. There was no significant difference

in the chemical composition of analyzed areas found by EDX, which indicates significant solubility of corrosion products [23].

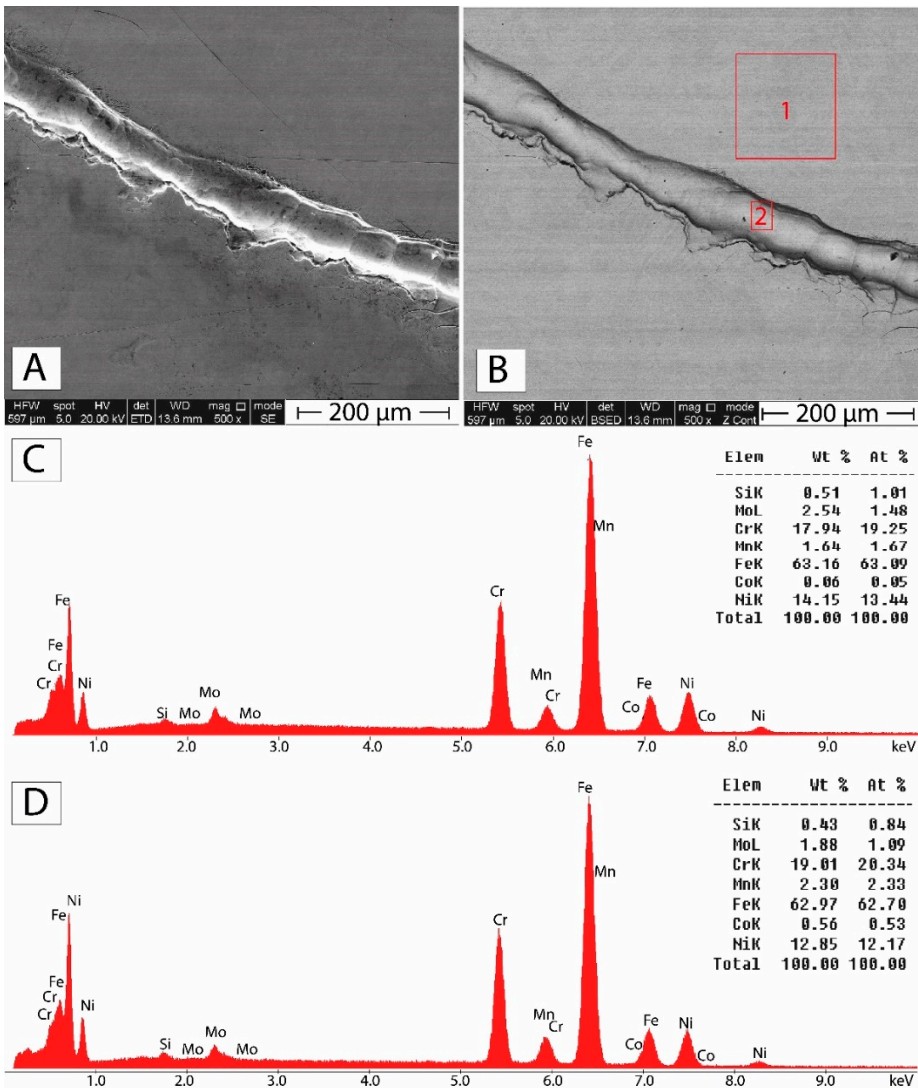

**Figure 6.** (**A**) Surface after corrosion test in SE, (**B**) surface after corrosion test in BSE with marked areas of EDX analysis, (**C**) EDX spectrum for area No.1, (**D**) EDX spectrum for area No.2.

## 4. Discussion

The metallic implant can release various elements into the surrounding tissues during their lifetime. If the concentration of released elements reaches critical doses, these elements can induce adverse effects around the area of implantation or in the whole body. Therefore, it is necessary to determine the potential toxicity of metal ions and metal-based particulates that may be released from medical implants. Medical implants are exposed to diverse conditions related to pH and the internal environment of the human body and consequently undergo various types of corrosion leading to the release of metal ions or wear particulates emission [24–26].

Although there are various methods of surface treatment of implants of stainless steel, the chemical passivation is both very simple and also one of the most effective, therefore it had previously been standardized according to ASTM or ISO for specific application in implantology [27]. During this procedure, a bath of oxidizing acid is used. In this study, a bath of 30% nitric acid was used. During the process of passivation, the chromium contained in the materials is primarily oxidized into the form of chromium trioxide which acts as a barrier on the free surface [28]. The composition of a passive layer formed on the

surface of the implant spontaneously if exposed to air is very similar to the passivation layer formed on the surface chemically, the main difference is its thickness. The passive layer usually has a thickness of several nm and can be spontaneously reformed on the surface even after electrochemical reduction or mechanical damage. On the other hand, the passivation layer can be up to 1 μm thick and, after its reduction or mechanical damage, can be reformed only by repeated chemical treatment [29]. The stability of the oxide-based protective layer, either passive or passivated, can be characterized by its breakdown potential during potentiodynamic polarization. During this test, the aggressivity of the environment corresponding to its electrochemical potential is supplemented by electric potential created on the sample by a potentiostat device [30]. According to ASTM and ISO standards, the electric potential is progressively increased, which simulates the increasing aggressivity of the corrosion environment [31].

The passivation layer is locally reduced electrochemically when breakdown potential in the transpassive region is reached during polarization accompanied by a steep increment of current density which indicates intensive corrosion of base material. There are two methods for the determination of breakdown potential, the first is mainly graphical as the coordinates of the inflection point are determined directly from the curve recordings [32,33]. It was suggested by [34–36] that pitting corrosion, as determined by zero resistance amperometry (ZRA), becomes stable above $10^{-5} A/cm^{-2}$. Thus, pitting potentials can also be determined as the potential at which the current density reaches $10^{-5} A/cm^{-2}$. It was found that previously corroded areas are repassivated by a more electrochemically stable layer as breakdown potential measured during the second polarization is shifted from 750 to 870 mV vs. SCE [37].

Once the potential is decreased again to the passive region, the damaged area is spontaneously repassivated, but this part of the curve was not captured in this particular experiment as the only influence of previous passivation layer damage to corrosion properties of the newly formed passive layer was investigated [38–40].

Although the main benefit of the potentiodynamic polarization test is related to information connected to breakdown potential obtained from the transpassive region, additional characteristics can also be obtained from the Taffel region located on the interface of immunity and passivity regions [41]. According to [42–44], for stainless steels in a passive state, there is a dynamic corrosion equilibrium set-up between the corrosion solution and oxide-based surface layer. The corrosion current density is a function of the amount of base material atoms being actively corroded underneath the barrier of the passive or passivation layer by elements penetrating through the layer. It was found that thicker, more dielectric, and compact layers protect the base material more effectively than thinner or less compact ones, thereby reducing the corrosion current density and hence corrosion rate at passive state [45,46]. These reactions took place across the whole exposed surface area, therefore there is an assumption that this process is closer to general corrosion rather than pitting corrosion which occurs mainly in the transpassive region when the oxide layer is severely damaged [47].

When results obtained from the Tafel slope region were compared, there were certain differences found between the first curve recorded on an "as-received" surface after chemical passivation and the second curve recorded on a surface previously corroded during the first polarization. As the areas activated and actively corroded during the first polarization were spontaneously repassivated, the data measured during the second polarization were significantly affected by the character of the passive layer newly formed on previously active areas where pitting corrosion occurred [48,49].

It was found that surfaces exposed to previous polarization show significant shifts of corrosion potential to more noble values when compared to as-received state (134 mV vs. 414 mV vs. SCE). This indicates that the passive layer formed on previously active areas is composed of more noble types of chromium-based oxides than the layer developed during chemical passivation. The influence of additional electrochemical passivation of areas remaining in the passive state during the first polarization was studied before [41] and may

also relate to the increase of corrosion potential. The shift of the corrosion potential to more noble values reduces the risk of coupled galvanic corrosion development in areas where the plate is in direct contact to, e.g., fixing screws made from Al-rich Ti alloys [50].

The reduction of polarization resistance from 107 k$\Omega \cdot$cm$^2$ for the passivated surface to 84 k$\Omega \cdot$cm$^2$ for repassivated surface indicates that the newly formed layer is probably thinner [51] and less dielectric [52] than the layer formed during passivation.

Corrosion rates were calculated from the corrosion current density evaluated from the Tafel slope region occurring in the passive state region [53,54]. It was found that the passivated surface exhibits a lower corrosion rate in comparison to the previously polarized and corroded surface (0.76 μm/year vs. 2.01 μm/year). The corrosion rate directly correlates to the amount of nickel ions directly released from areas of corrosion in form of water-soluble complexes [39,55] and can reach values up to 263.2 ng/year per exposed square centimeter of the application. The amount of other ions (Cr, Mn, Mo) released from the material was also increased for the second polarization cycle. It has been determined that $1 \times 10^{12}$ particles/mL and Co$^{2+}$ ions at 1000 μM (589 ppm) concentration cause numerous levels of toxicity to macrophages and Cr$^{3+}$ did not exhibit toxicity at the same concentration. Contrary to that, another study reported that Co$^{2+}$ and Cr$^{3+}$ are toxic at concentrations 8–10 ppm and 350–500 ppm, respectively. Moreover, size-dependent toxicity has not been fully studied yet. These differences in reported toxicities are the result of multivariate research methodologies which are being used within this field and show the importance of establishing new and more appropriate methodologies [56]. There were no solid corrosion products detected on the surface after the corrosion test, this indicates the fact that all corrosion products were dissolved into corrosion solution in form of ions. If this kind of massive damage occurs spontaneously in the body, all the ions will be absorbed by surrounding tissue. Some of them, consisting mostly of biogenic elements will be metabolized during the corrosion process causing only minimal damage. On the other hand, corrosion products based on Cr, Ni, or Mo ions will be simultaneously accumulated in surrounding tissue or spread throughout the body while accumulating in the liver and kidneys [57,58]. The rounded shape of the corroded area indicates that there was a crevice mechanism preferred rather than pitting. This indicates high stability of the surface layer which was terminally damaged in areas affected by galvanic coupling between areas of different oxygen levels while other exposed areas remained intact [59]. There were no solid-state oxide-based corrosion products found on the surface after polarization. This means that all metallic ions released during corrosion formed water soluble compounds [60].

Due to the risks mentioned above, the applications from 1.4441 steel are primarily intended only for short- to mid-term implantations followed by their explantation. In contrast with long-term titanium implants, where strong bonding to hard tissue is elementary, only weak or even no bonding to fixed bones is required in this case in order to avoid possible damage during the explantation process. The character of bonding with respect to interaction between implants and surrounding tissue is determined, inter alia, by the wettability of implant free surface represented by its contact angle. Previous research indicates that surfaces with low contact angles close to 0° primarily induce the growth of hard tissue and adhere bone cells effectively; on the other hand, surfaces with high contact angle up to 80° induce growth of soft fibrous connective tissue which is preferred in case of short-term implant intended for explantation [61]. The wettability analysis performed in different areas of the implant showed consistent results. The contact angles for areas near and further from threaded holes varied insignificantly with an average value of 50° respective 52°. Therefore, it can be predicted that there will be a partial bonding between hard tissues and the implant, but most of the bond will be formed by soft tissue, dominantly consisting of fibrous cells [62].

## 5. Conclusions and Outlooks

In the present work, the multiple characteristics of the anterolateral distal tibial plate from AISI 1.4441 stainless steel were evaluated. It was found that chemical passivation

of the tested implant surface forms a surface layer with moderate wettability represented by a contact angle of approximately 52°. This suggests that there will be predominantly soft fibrous tissue formed and bonded to the implants interfaces which will enable its safe resection if needed. Corrosion parameters of the passivated surface were measured by potentiodynamic polarization when 2 voltammetry cycles were performed. The first cycle was applied to determine the characteristics of the surface layer after the chemical passivation. The first cycle was stopped when the stable pitting occurred simulating real damage of the surface under extreme conditions. The second cycle of polarization was applied on the same surface to characterize the surface ability for self-passivation. It was found that surface after chemical passivation showed less noble corrosion potential, but its corrosion rate was approximately three times lower. Although the values of the corrosion rate are insignificant and will probably not cause any functional problems even after long-term exposure, the amount of nickel and other ions released during the corrosion process may cause pathological changes or even rejection of the implant. According to the outcomes of the presented work, the risk of fatal implant failure significantly increases if its surface is damaged by pitting corrosion during its lifetime.

**Author Contributions:** Conceptualization, J.H.; methodology, J.H. and K.P.D.; validation, J.H. and K.D.; formal analysis, J.H. and R.M.; investigation, J.H., K.D. and P.S.; resources J.K. and K.F.; data curation, J.H. and T.E.L.D., writing—original draft preparation, J.H., K.D. and K.P.D.; writing— J.H., T.E.L.D. and J.K.; visualization, J.H.; supervision, J.H.; project administration, K.P.D.; funding acquisition, J.K. All authors have read and agreed to the published version of the manuscript.

**Funding:** This paper was created with the contribution of the following projects "No. CZ.02.1.01/0.0/ 0.0/17_049/0008441 Innovative therapeutic methods of the musculoskeletal system in accident surgery within the Operational Programme Research, Development and Education financed by the European Union and from the state budget of the Czech Republic" and "Student Grant Competition SP2021/48 Optimization of the properties of advanced technical materials by control of their microstructural parameters".

**Institutional Review Board Statement:** Not applicable.

**Informed Consent Statement:** Not applicable.

**Data Availability Statement:** Data available in a publicly accessible repository that does not issue DOIs. Publicly available datasets were analyzed in this study. This data can be found here: [https: //fastshare.cz/19260857/upload-metals-hlinka.rar] (accessed on 29 September 2021).

**Conflicts of Interest:** The authors declare no conflict of interest.

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
