# Peer review of "Complex Material and Surface Analysis of Anterolateral Distal Tibial Plate of 1.4441 Steel"

_metals, doi:10.3390/met12010060_

Round 1

Reviewer 1 Report

This manuscript investigates the microstructure and microhardness changes near threaded holes of anterolateral distal tibial plate as well as their corrosion performance. One aspect should be explained befor published in this journal.

What is the relationship between the corrosion performance and the surface film properties? It is better for the authors to discuss this probelm.

Author Response

Dear reviewer.

Many thanks for your time spend on this revision. We have done our best with final changes. There were a large sections added mainly to “Results” and “Discussion” related to surface aspect, resp. how the properties of layer formed on the surface after chemical passivation resp. spontaneous passivation affects final corrosion properties. Also, other details connected to corrosion test itself were added to the manuscript.

I how this will improve the quality of the paper it will become even more.

Reviewer 2 Report

This paper's purpose was the investigation of the surface properties (microstructure, corrosion, wettability, microhardness, and contact-type surface roughness test) of an implant, intended for ankle reparation, made of AISI 1.4441. As this nickel-rich austenitic steel is widely used to manufacture mainly short-term hard tissue reinforcements, complications related to ions releasing during specific corrosion processes may result in its terminal failure. Based on this supposed complication, the motivation of research was to primarily evaluate the number of ions released from the application under conditions simulating its real use. The paper is well written, but some typos were found. The subject to be studied is well presented in the introduction, with current and adequate references (around 42% in the last five years). The methods are clearly presented and are the classical methodology for preparing and characterizing this class of materials. The results are interesting, and their analysis is adequate, based on adequate earlier findings. The conclusions are included in the discussion section. I suggest their separation, with the conclusion as a new section. This paper may add very interesting knowledge for biomedical metals applications. I suggest its publication after considering the points above

Author Response

Many thanks for your time spend on this revision. We have done our best with final changes. There were a large sections added mainly to “Results” and “Discussion” related to surface aspect, resp. how the properties of layer formed on the surface after chemical passivation resp. spontaneous passivation affects final corrosion properties. Also, other details connected to corrosion test itself were added to the manuscript.

To the separation of "Conclusions" from "Discussion"-I'm not sure about this. The findings and conclusions outcoming from this research are fairly complex and I believe it is more useful for the reader to have them followed by proper explanation. But if you really insists on his, I'm ready do separate them.

Reviewer 3 Report

The authors should ask the support of an English language expert to make their paper easier to read. This, unfortunately, is an important issue of tha paper and makes its evaluation more complex.

On the scientific point of view it is not clear how the amount of cations released per year is measure having performed two scans in sequence without a test of weight change in function of dwell time in the physiological solution. This would have helped as well to have surfaces suitable for SEM observation of the oxidized layer eventually formed on top of the metal. 
The analysis of the deeping solution moreover would have supported the weight change data and better supported both discussion and conclusions

Author Response

Dear reviewer.

Many thanks for your time spend on this revision. We have done our best with final changes. There were a large sections added mainly to “Results” and “Discussion” related to surface aspect, resp. how the properties of layer formed on the surface after chemical passivation resp. spontaneous passivation affects final corrosion properties. Also, other details connected to corrosion test itself were added to the manuscript. We have also incorporated your recommendation into the manuscript and corrected the imperfections you have pointed out-I would like to thank you for this!

To your point connected to material ions leaching, resp. weight changes of tested metal plate-

The standards used for the corrosion tests (ASTM F 2129, ASTM G 61) only uses the polarization method for corrosion properties determination. The leaching of each element is serious problem and is measured by long term leaching test (ISO 10993-15), other than used in this case. If the amount of the ions released from the material into the solution will be measured after polarization tests, its amounts would be literally gargantuan. There was no point of investigation of each element’s concentration in the corrosion solution as the tests simulates the highly accelerated corrosion damage (especially if the break-down potential if reaches-and it was during the test…), the concentration of the elements will definitely not corelate to the time of its exposure in the solution. The leaching tests was used in other experiments of our team which will be hopefully published soon-it was performed by implant’s immersion for 7 days period followed by spectroscopic analysis of Ni and other elements concentration in the solution…

The main reason why the polarization tests is basically used for this type of implant is that even 30 days is not enough time to find statistically significant differences in weight of the sample. Anyway, the method used for this implants if fully standardized and even considered by international authorities as fully sufficient for corrosion rates evaluation.

Reviewer 4 Report

The paper discuss about the surface properties of a surgical stainless steel regarding the possibility of "in vivo" toxic metal ions release. In principle the work is well presented and the experiments correctly conceived and carried out. There are several typo mistake and imprecisions that should be amended. 

There is also a technical issue I would discuss.

below the list of my concerns:

The paper focus on the possibile ion release from surgical stainless steel. The evaluation of the ions release rate  was theoretically calculated (table 5 and table 6). Why the authors did not take into account possible selective leaching for the single alloys components? that's could tremendously affect the material biocompatibility. Please asses this point.

minor issues

page 2 12th row  dot is missing"...implantantion."

page 2 : The se mechanism ... !? Please explain.

page 4 ... by the SEE system ..."!? Please explain

page 4 paragraph 3. results. figure 2A and 2B instead of fig 1A and  1B. 

page 6, figure 3A and 3B instead of fig 2A and  2B.

Table 3. Too many digits are reported for the contact angle easurement. The  uncertinity is always larger than 1° making the use of four digits a non sense. Furthermore the uncertainity is larger than the difference between the two samples making them statistically not distinguishable.

Table 4. Number of digits much larger than the experimental precision of the measurements.

Bottom of page 8. "...atomic volume fractures of manganese..." 

Tables 5 and 6 displays the same data in different units (ng/year cm2 and nmol/year cm2). That's unecessary redundant. The tables can be merged if not reduced to only one.

page 10 . 11th row. "...direct..." instead of "dirrect"

Author Response

Dear reviewer.

Many thanks for your time spend on this revision. We have done our best with final changes. There were a large sections added mainly to “Results” and “Discussion” related to surface aspect, resp. how the properties of layer formed on the surface after chemical passivation resp. spontaneous passivation affects final corrosion properties. Also, other details connected to corrosion test itself were added to the manuscript. We have also incorporated your recommendation into the manuscript and corrected the mistakes you have point out-I would like to thank you for this!

To the separation of "Conclusions" from "Disscussion"-I'm not sure about this. The findings and conclusions outcomming from this research are fairly complex and I believe it is more seful for the reader to have them followed by proper explanation. But if you realy insists on his, I'm ready do separate them.

To your point connected to material ions leaching-

The standards used for the corrosion tests (ASTM F 2129, ASTM G 61) only uses the polarization method for corrosion properties determination. The leaching of each element is serious problem and is measured by long term leaching test (ISO 10993-15), other than used in this case. If the amount of the ions released from the material into the solution will be measured after polarization tests, its amounts would be literally gargantuan. There was no point of investigation of each element’s concentration in the corrosion solution as the tests simulates the highly accelerated corrosion damage (especially if the break-down potential if reaches-and it was during the test…), the concentration of the elements will definitely not corelate to the time of its exposure in the solution. The leaching tests was used in other experiments of our team-it was performed by implant’s immersion for 7 days period followed by spectroscopic analysis of Ni and other elements concentration in the solution…

Round 2

Reviewer 3 Report

The paper resubmitted presents a real improvement in the text quality but it needs a further improvement of the English language.

The table listing the steel alloy composition is indicated as At.% which is doubtful. The authors should carefully check and, in case, correct or justify.

The data seem more consistent however it the paper has a wide discussion but no conclusions. The authors are demanded to summarize the most important conclusions in a separate chapter to increase the paper readability.

Author Response

Dear Reviewer,

Thank you for your comments and time to help us to make our paper more attractive for readers.

We had considered your suggestions and we did some fine changes in our manuscript as followed:

  • The manuscript was proofread by one of the native English-speaking co-authors from the UK, so we believe the English quality improved significantly.
  • You were right about the composition, we checked it and it was corrected to Wt.%, thank you for this!
  • We had summarized the most significant outlooks and added chapter “discussion and outcomes” into the manuscript, where the most significant findings are mentioned.

I believe the quality of the manuscript now matches the journal criteria and will be beneficial for its readers.

Thank you

Regards,

Josef Hlinka
